# Adipose Triglyceride Lipase Deficiency Attenuates In Vitro Thrombus Formation without Affecting Platelet Activation and Bleeding In Vivo

**DOI:** 10.3390/cells11050850

**Published:** 2022-03-01

**Authors:** Madeleine Goeritzer, Stefanie Schlager, Katharina B. Kuentzel, Nemanja Vujić, Melanie Korbelius, Silvia Rainer, Dagmar Kolb, Marion Mussbacher, Manuel Salzmann, Waltraud C. Schrottmaier, Alice Assinger, Axel Schlagenhauf, Corina T. Madreiter-Sokolowski, Sandra Blass, Thomas O. Eichmann, Wolfgang F. Graier, Dagmar Kratky

**Affiliations:** 1Molecular Biology and Biochemistry, Gottfried Schatz Research Center, Medical University of Graz, 8010 Graz, Austria; madeleine.goeritzer@gmx.at (M.G.); schlager.stefanie@gmail.com (S.S.); katharina.kuentzel@medunigraz.at (K.B.K.); nemanja.vujic@medunigraz.at (N.V.); m.korbelius@medunigraz.at (M.K.); silvia.rainer@medunigraz.at (S.R.); corina.madreiter@medunigraz.at (C.T.M.-S.); sandra.blass@medunigraz.at (S.B.); wolfgang.graier@medunigraz.at (W.F.G.); 2AOP Orphan Pharmaceuticals GmbH, 1190 Vienna, Austria; 3Core Facility Ultrastructural Analysis, Medical University of Graz, 8010 Graz, Austria; dagmar.kolb@medunigraz.at; 4BioTechMed-Graz, 8010 Graz, Austria; thomas.eichmann@medunigraz.at; 5Department of Pharmacology and Toxicology, University of Graz, 8010 Graz, Austria; marion.mussbacher@uni-graz.at; 6Department of Internal Medicine II/Cardiology, Medical University of Vienna, 1190 Vienna, Austria; manuel.salzmann@meduniwien.ac.at; 7Institute of Vascular Biology and Thrombosis Research, Center for Physiology and Pharmacology, Medical University of Vienna, 1190 Vienna, Austria; waltraud.schrottmaier@meduniwien.ac.at (W.C.S.); alice.assinger@meduniwien.ac.at (A.A.); 8Department of General Pediatrics and Adolescent Medicine, Medical University of Graz, 8010 Graz, Austria; axel.schlagenhauf@medunigraz.at; 9Institute of Molecular Biosciences, University of Graz, 8010 Graz, Austria; 10Core Facility Mass Spectrometry, Medical University of Graz, 8010 Graz, Austria

**Keywords:** platelets, adipose triglyceride lipase, thrombosis, mitochondrial respiration, intravital microscopy

## Abstract

According to genome-wide RNA sequencing data from human and mouse platelets, adipose triglyceride lipase (ATGL), the main lipase catalyzing triglyceride (TG) hydrolysis in cytosolic lipid droplets (LD) at neutral pH, is expressed in platelets. Currently, it is elusive to whether common lipolytic enzymes are involved in the degradation of TG in platelets. Since the consequences of ATGL deficiency in platelets are unknown, we used whole-body and platelet-specific (plat)Atgl-deficient (−/−) mice to investigate the loss of ATGL on platelet function. Our results showed that platelets accumulate only a few LD due to lack of ATGL. Stimulation with platelet-activating agonists resulted in comparable platelet activation in Atgl−/−, platAtgl−/−, and wild-type mice. Measurement of mitochondrial respiration revealed a decreased oxygen consumption rate in platelets from Atgl−/− but not from platAtgl−/− mice. Of note, global loss of ATGL was associated with an anti-thrombogenic phenotype, which was evident by reduced thrombus formation in collagen-coated channels in vitro despite unchanged bleeding and occlusion times in vivo. We conclude that genetic deletion of ATGL affects collagen-induced thrombosis without pathological bleeding and platelet activation.

## 1. Introduction

Platelets are small anucleated blood cells, which are derived from megakaryocytes in the bone marrow. In addition to their well-established classical role in thrombosis and hemostasis, platelets have important immunoregulatory functions. After activation and degranulation, platelets release numerous mediators that lead to the recruitment of leukocytes and progenitor cells to sites of vascular injury and inflammation. In this way, platelets are involved in the etiology of various diseases, such as diabetes, atherothrombosis, cardiovascular and autoimmune diseases [1,2,3,4]. 

While platelets circulate freely under physiological conditions, endothelial injury leads to the exposure of subendothelial extracellular matrix proteins and triggers platelet activation via a multi-step process [5]. Platelet tethering and adhesion to the endothelial wall is mediated by multiple interactions between platelet membrane receptors (integrins and glycoproteins) and various ligands in the damaged endothelium, particularly von Willebrand Factor (vWF), fibrinogen, and collagen. Platelet activation is amplified by soluble agonists such as adenosine diphosphate (ADP), thromboxane A2 (TXA2), or thrombin, which mediate their action via G-protein-coupled receptors [6]. Collectively, these events provide a stimulus for the conformational change of platelet αIIbβ3 integrin (GPIIb/IIIa receptor), which (in its activated form) can bind fibrinogen and vWF, thereby triggering platelet aggregation and consequently stable thrombus formation [6,7]. 

A number of pathophysiological conditions related to dyslipidemia, including atherosclerosis, diabetes, and the metabolic syndrome, have been associated with increased platelet reactivity and thrombogenic potential [8]. Hyperlipidemia, especially hypercholesterolemia, primes platelets and increases platelet reactivity in response to various agonists [9]. In patients, hypercholesterolemia is associated with increased platelet activity, such as hyperaggregability [10,11], and the administration of lipid lowering drugs reverses this prothrombotic phenotype [12]. Patients with Tangier’s disease, a disorder that is characterized by the virtual absence of high-density lipoproteins (HDL), show defective activation in response to classical platelet agonists [13]. The sterol composition of platelet membranes affects their fluidity, organization, and physiological activity [14]. Platelets are composed of phospholipids (PL), sterols, sphingolipids, fatty acids, and glycerolipids, and contain appreciable amounts of triglycerides (TG) and cholesteryl esters (CE) [4,15,16]. During activation, bioactive lipids (e.g., 1,2-diacylglycerol (DG), fatty acids (FA), eicosanoids, phosphatidylinositides, lysophospholipids) are formed and participate in the regulation of numerous cellular processes by modulating platelet membrane composition. The availability of FA is regulated by the action of lipid hydrolases via breakdown of intracellular lipid stores. CE degradation is catalyzed by CE hydrolases to generate cholesterol and FA. In addition, FA are generated by neutral lipases, which sequentially hydrolyze TG stored in cytosolic lipid droplets (LD) at neutral pH. The first step of neutral lipolysis is mediated by adipose triglyceride lipase (ATGL) to generate DG and FA. Hormone-sensitive lipase (HSL) degrades DG to release the second FA and monoacylglycerol. Finally, monoglyceride lipase (MGL) hydrolyzes monoacylglycerol to release FA and glycerol [15]. 

The importance of neutral lipolysis became evident by the severe phenotype of cardiac myopathy in humans [16] and mice [17,18] affected by ATGL deficiency. The lipolytic defect in Atgl-deficient (−/−) mice leads to TG accumulation in various cell types and tissues. Studies by our group have shown that loss of ATGL in macrophages results in TG-rich LD accumulation and phagocytic defects [19], impaired macrophage migration [20], ER stress and apoptosis, leading to mitochondrial dysfunction [21,22]. Additionally, ATGL deficiency in myeloid cells on the low-density lipoprotein receptor-deficient background resulted in reduced atherosclerotic plaque formation [23]. Moreover, the release of lipid mediators in neutrophils depends on the hydrolysis of the TG-rich pool of LD by ATGL [24]. Here, we hypothesized that ATGL deficiency may also affect platelet function. 

## 2. Materials and Methods

### 2.1. Animals

Global Atgl−/− mice were generated as described elsewhere [18]. Mice with a targeted deletion of *Atgl* in platelets (platAtgl−/− mice) were obtained by crossing Atgl^flox/flox^ mice (kindly provided by Erin Kershaw, University of Pittsburgh, Pittsburgh, PA, USA) with transgenic mice that express Cre recombinase under the control of the platelet factor 4 promoter (C57BL/6-Tg (Pf4-icre) Q3Rsko/J; Pf4 Cre; C57BL/6 background; kindly provided by Bernhard Nieswandt, University of Würzburg, Würzburg, Germany). Wild-type littermates (wt) and Atgl^flox/flox^ mice, respectively, were used as controls. Mice were fed a standard chow diet (4% fat and 19% protein; Altromin Spezialfutter GmbH & Co, Lage, Germany) and water ad libitum on a regular light–dark cycle (12 h light, 12 h dark). All protocols were approved by the Austrian Federal Ministry of Science, Research and Economy, Division of Genetic Engineering and Animal Experiments, Vienna, Austria (BMBWF-66.010/0165-V/3b/2019, BMWFW-66.010/0197-WF/V/3b/2017, BMWF-66.010/0153-WF/V/3b/2015). 

### 2.2. Megakaryocyte Isolation and Differentiation

Bone marrow-derived mature megakaryocytes were generated as described [25]. Briefly, mouse femurs were flushed, and cells expressing Ly6G/Ly6C, CD11b, CD16/32, and B220 were depleted using magnetic beads (sheep anti-rat IgG Dynabeads, Thermo Fisher Scientific, Waltham, MA, USA) and the following antibodies: anti-mouse Ly6G/Ly6C (561103, BD Biosciences, Franklin Lakes, NJ, USA), anti-mouse CD11b (47-0112-82, Thermo Fisher Scientific), anti-mouse CD16/CD32 (553141, BD Biosciences), and anti-mouse B220 (BD Biosciences). Negatively selected cells were incubated in megakaryocyte medium (Stempro-34 SFM, Thermo Fisher Scientific) containing 2.6% nutrient supplement, 1% glutamine, 1% penicillin–streptomycin–fungizone, and 20 ng/mL stem cell factor (Peprotech EC Ltd., London, UK) for 2 d at 37 °C and 5% CO_2_, followed by a 5-d incubation with additional 50 ng/mL thrombopoietin (TPO) (Peprotech EC Ltd.). Mature megakaryocytes were enriched with a gradient of 3%/1.5% BSA (PAA Laboratories, Fisher Scientific, Hampton, NH, USA) under gravity for 45 min at RT. Cells in the lower 25% of the gradient, representing mature megakaryocytes, were washed in PBS and harvested in TriFast™ (VWR, Radnor, PA, USA). 

### 2.3. Platelet Isolation and Purification

For RNA, Western blotting, and lipid analyses, mouse platelets were isolated and purified as described [26]. In brief, blood was collected from the retrobulbar plexus, anticoagulated with 3.8% sodium citrate, and centrifuged at 400× *g* for 20 min to obtain platelet-rich plasma (PRP). Washed platelets were isolated from the PRP by centrifugation (10 min, 1300× *g*), and platelet pellets were resuspended in Tyrode’s buffer (140 mM NaCl, 3 mM KCl, 1 mM MgCl_2_, 16.63 mM NaHCO_3_, 10 mM HEPES, pH 7.4). 

Platelets for the measurements of oxygen consumption rate (OCR) were isolated from 440 µL of blood with 60 µL anticoagulant citrate–dextrose (ACD) buffer (Sigma-Aldrich, St. Louis, MO, USA) and 300 µL Tyrode’s buffer. Blood was centrifuged at 200× *g* for 6 min, and the plasma and ~1/3 of the red fraction were transferred to a new tube. The samples were centrifuged at 100× *g* for 6 min in a swing-out rotor, and the upper layer containing platelets was transferred to a new tube. An additional 200 µL of Tyrode’s buffer was added to the remaining red fraction, inverted, and centrifuged again at 100× *g* for 6 min in a swing-out rotor. The upper layers containing platelets were pooled and mixed with 1/25 volume of ACD and 1/100 volume of apyrase (50 U/mL; Sigma-Aldrich) to inhibit preactivation.

To purify platelets, cells were incubated with anti-Ter-119 and anti-CD45 beads (Miltenyi Biotec, Bergisch Gladbach, Germany) to remove residual red blood cells and leukocytes, respectively. 

### 2.4. RNA Isolation and Quantitative Real-Time PCR Analysis

RNA from megakaryocytes was isolated using TriFast™ reagent according to the manufacturer’s protocol (VWR, Radnor, PA, USA). One microgram of total RNA was reverse transcribed using the High Capacity cDNA Reverse Transcription Kit (Thermo Fisher Scientific, Waltham, MA, USA) according to the manufacturer’s instructions. Quantitative real-time PCR was performed on a Bio-Rad CF X96 Real-Time System (Bio-Rad, Hercules, CA, USA) using the GoTaq^®^ qPCR Mastermix (Promega, Madison, WI, USA). Samples were analyzed in duplicate and normalized to hypoxanthine–guanine phosphoribosyltransferase (*Hprt*) expression as the reference gene. Expression profiles and associated statistical parameters were determined by the 2^−ΔΔCT^ method. 

Total RNA from isolated platelets was extracted using TriFast™ reagent (VWR) according to the manufacturer’s protocol. Five hundred nanograms of total RNA was reverse transcribed using the High Capacity cDNA Reverse Transcription Kit (Thermo Fisher Scientific). PCR products were analyzed by agarose gel electrophoresis. 

Specific oligonucleotide primers were designed as follows: *Hprt*, fwd 5′-TCAGTCAACGGGGGACATAAA-3′, rev 5′-GGGGCTGTACTGCTTAACCAG-3′, *Atgl*, fwd 5′-GCCACTCACATCTACGGAGC-3′, rev 5′-CCACGGATGGTGTTC-3′; *Hsl*, fwd 5′-GCTGGTGACACTCGCAGAAG-3′, rev 5′-TGGCTGGTGTCTCTGTGTCC-3′; *Mgl*, fwd 5′-CGGACTTCCAAGTTTTTGTCAGA-3′, rev 5′-GCAGCCACTAGGATGGAGATG-3′; *Cd41*, fwd 5′-TTCTTGGGTCCTAGTGCTGTT-3′, rev 5′-CGCTTCCATGTTTGTCCTTATGA-3′; *Cd45*, fwd 5′-ATATCGCGGTGTAAAACTCGTC-3′, rev 5′-TAGGCTTAGGCGTTTCTGGAA-3′; *Cd235a*, fwd 5′-GGTAACCCAAATCAGCATTCAGC-3′, rev 5′-GGTGACGGCATTCCTCCAA-3′, *Cgi-58*, fwd 5′-GGTTAAGTCTAGTGCAGC-3′, rev 5′-AAGCTGTCTCACCACTTG-3′, *G0S2*, fwd 5′-GTGAAGCTATACGTGCTGGG-3′, rev 5′-CCGTCTCAACTAGGCCGAG-3′, *Hilpda*, fwd 5′-TGCTGGGCATCATGTTGACC-3′, rev 5′-TGACCCCTCGTGATCCAGG-3′.

### 2.5. Western Blotting Analysis

Purified platelets were isolated from a pool of blood from 10 wt mice as described above. Protein lysate (10 µg) from white adipose tissue (WAT) of wt mice was used as a control. Gonadal WAT was surgically removed, washed in ice-cold PBS containing 1 mM EDTA and homogenized on ice in lysis buffer A (0.25 M sucrose, 1 mM EDTA, 1 mM DTT, pH 7.0) using an ultra-turrax^®^ (IKA-Werke GmbH & Co. KG, Staufen, Germany). The infranatants were collected after centrifugation at 20,000× *g* and 4 °C for 60 min. Proteins were separated under reducing conditions (25 mM DTT) by SDS-PAGE and transferred to a nitrocellulose membrane (Hybond-C Extra; GE Healthcare, Chicago, IL, USA). Non-specific binding sites were blocked by incubating the membrane with 5% non-fat dry milk (Sigma-Aldrich, St. Louis, MO, USA) in 1× TBS-T buffer (Tris-buffered saline with Tween20) for 1 h at RT. Immunodetection was performed using mouse anti-ATGL (1:200, #2138S, Cell Signaling Technology, Danvers, MA, USA), mouse anti-HSL (1:800, #4107, Cell Signaling Technology), and rabbit anti-MGL (1:1000; kindly provided by Robert Zimmermann, University of Graz, Graz, Austria) [27]. Horseradish peroxidase-conjugated goat anti-rabbit (1:5000) and rabbit anti-mouse antibodies (1:1000) (Dako, Glostrup, Denmark) were visualized by enhanced chemiluminescence detection (ECL Plus; Thermo Fischer Scientific, Waltham, MA, USA) on an X-ray film.

### 2.6. Electron Microscopy

Washed platelets were collected in cellulose capillaries, fixed in 2% paraformaldehyde/2.5% glutaraldehyde (*w*/*v*) for 1 h at RT, washed, post-fixed in cacodylate buffer/2% osmium tetroxide (*w*/*v*) for 1 h, and washed with cacodylate buffer. After dehydration in graded series of ethanol, platelets were infiltrated (propylene oxide and TAAB embedding resin, pure TAAB embedding resin) for 3 h, placed in TAAB embedding resin (2 × 90 min), transferred into embedding molds, and polymerized (72 h, 60 °C). Ultrathin sections (75 nm) (Leica Microsystems, Wetzlar, Germany) were stained with lead citrate (5 min) and Platinum Blue (International Bio-Analytical Industries, Inc., Boca Raton, FL, USA) (15 min). Images were taken at 120 kV using a FEI Tecnai G2 20 transmission electron microscope (FEI Corporation, Eindhoven, The Netherlands) with a Gatan ultrascan 1000 CCD camera. 

For scanning electron microscopy, washed platelet suspensions (50 µL) were mounted on cover slips and immediately fixed with 2% paraformaldehyde/2.5% glutaraldehyde (*w*/*v*) for 1 h. Samples were post-fixed in 2% osmium tetroxide (*w*/*v*) for 1 h at RT and dehydrated in graded series of ethanol (30–96% and 100% (*v*/*v*)). The cover slips were treated with 1,1,1,3,3,3-hexamethyldisilazane for 5 min and air dried before being placed on stubs covered with a conductive double-coated carbon tape. Images were taken using a Sigma 500VP FE-SEM with an SEM detector (Carl Zeiss, Oberkochen, Germany) operated at an acceleration voltage of 3 kV. 

### 2.7. Targeted Lipidomic Analysis

Protein content was quantified by flow cytometry (Guava^®^ easyCyte^TM^ 8, Merck KGaA, Darmstadt, Germany), cell pellets (in 140 µL dH_2_O) were transferred to 2 mL Safe-Lock PP tubes, and lipids were extracted according to Matyash et al. [28]. In brief, samples were homogenized with two 6 mm steal beads in a mixer mill (Retsch, Haan, Germany; 2 × 10 s, frequency 30/s) in 700 µL methyl tert-butyl ether/methanol (3/1, *v*/*v*) containing 500 pmol butylated hydroxytoluene, 1% acetic acid, and 200 pmol of internal standards (TG 45:0, PC 34:0, PE 34:0, PS 34:0; Avanti Polar Lipids, Alabaster, AL and Larodan Fine Chemicals, Solna, Sweden) per sample. Total lipids were extracted under constant shaking for 45 min at RT. For phase separation, samples were centrifuged at 1000× *g* for 10 min. Thereafter, 500 µL of the upper, organic phase was collected and dried under a stream of nitrogen. Lipids were resolved in 150 µL of 2-propanol/methanol/H_2_O (7/2.5/1, *v*/*v*/*v*) for UPLC-MS analysis. 

Chromatographic separation was modified according to Knittelfelder et al. [29] using an ACQUITY-UPLC system (Waters Corporation, Milford, MA, USA) equipped with a Luna EvoC18 column (2.1 × 50 mm, 1.6 µm; Phenomenex, Torrance, CA, USA) starting with a 20 min linear gradient with 80% solvent A (methanol/H_2_O, 1/1, *v*/*v*; 10 mM ammonium acetate, 0.1% formic acid, 8 µM phosphoric acid). The column compartment was kept at 50 °C. An EVOQ Elite™ triple quadrupole mass spectrometer (Bruker, Billerica, MA, USA) with an ESI source was used to detect lipids in positive ionization mode. Lipid species were analyzed by selected reaction monitoring, and data were acquired using MS Workstation (Bruker). Data were normalized for recovery, extraction, and ionization efficacy by calculating analyte/internal standard ratios (AU) and normalizing to cell number.

### 2.8. Flow Cytometric Analyses of P-Selectin and αIIbβ3 Expression

Heparinized blood was diluted 1:5 with Tyrode’s buffer containing 2 mM CaCl_2_ (THCa). Four microliters of the diluted blood were mixed with 1 µL of agonists (ADP, 50 µM (Sigma-Aldrich, St. Louis, MO, USA); protease-activated receptor 4 agonist peptide AYPGKF-NH2 (PAR4-AP), 75 µM (Anaspec, Seraing, Belgium); cross-linked collagen-related peptide-XL (CRP), 10 µg/mL (kindly provided by Richard Farndale, Department of Biochemistry, University of Cambridge, Cambridge, UK); convulxin (CVX), 125 ng/mL (Santa Cruz, Heidelberg, Germany)) in a V-bottom 96-well plate. Blood cells were activated for 15 min at RT in the dark. Five microliters of a master mix containing anti-mouse CD41-FITC (1:200, 133903, Biolegend, San Diego, CA, USA), CD62P-PE-Cy7 (1:200, 148310, Biolegend; specific for P-selectin), and JON/A-PE (1:20, M023-2, directed against the activated form of mouse integrin αIIbβ3; Emfret Analytics, Wuerzburg, Germany) antibodies in THCa were added per 96-well, gently resuspended, and the cells were stained for 15 min at RT in the dark. Reactions were stopped by adding 190 µL 1% PFA in PBS. Samples (10,000 platelets) were analyzed by flow cytometry (Cytoflex S, Beckman Coulter Inc., Pasadena, CA, USA).

### 2.9. GPVI Staining

Blood was diluted (1:15) in Tyrode’s buffer and incubated with FITC-labeled anti-GPVI (JAQ1, Emfret Analytics, Wuerzburg, Germany) at saturating concentrations in the absence and presence of CRP (1 µg and 10 µg) for 15 min at RT. Cells were fixed and analyzed by flow cytometry (CytoflexS, Beckman Coulter Inc., Pasadena, CA, USA). CD41-APC (17-0411-80, eBioscience, Thermo Fisher Scientific, Waltham, MA, USA) was used as a platelet-specific marker. 

### 2.10. Platelet Aggregation Assay

Whole blood was collected from the retrobulbar plexus using a heparinized glass capillary and anticoagulated with 25 U/mL heparin/TBS. Aggregation was measured on a Multiplate^®^ analyzer (Multiplate Services GmbH, Munich, Germany). Samples were transferred to measurement wells containing 0.9% NaCl, and aggregation was induced by addition of collagen (final concentration 3.2 μg/mL, Hyphen Biomed, Neuville-sur-Oise, France). 

### 2.11. Tail Bleeding and Hemoglobin Assays

Bleeding assays were performed as described elsewhere [30]. Briefly, mice were weighed, anesthetized with ketamine and xylazine, placed in a prone position, and a 5 mm segment of the tail was amputated with a scalpel. The tail was immediately immersed in a vertical position in a 50 mL falcon tube containing pre-warmed 0.9% NaCl. Each mouse was monitored for 20 min, even when bleeding stopped, to detect rebleeding. Bleeding time was determined using a stopwatch. When bleeding on/off cycles occurred, the sum of bleeding times within the 20 min period was used. Body weight, including the tail tip, was measured again to determine the volume of blood loss by calculating the reduction in body weight. 

To confirm the accuracy of the determined bleeding volume caused by changes in body weight, we also performed a hemoglobin assay. Blood cells were separated by centrifugation at 1700× *g* for 5 min at RT. The supernatant was removed, and the erythrocytes were resuspended in 2 mL erythrocyte lysis buffer (ACK buffer) (Sigma-Aldrich, St. Louis, MO, USA) and incubated for 10 min at RT. The tubes were centrifuged at 10,800 × *g* for 5 min, and hemoglobin concentrations were measured spectrophotometrically at 540 nm using a Victor 1420 multilabel counter (PerkinElmer Life Sciences, Turku, Finland).

### 2.12. Mitochondrial Respiration Measurement 

Ten million platelets per well were seeded in an Agilent Seahorse XF96 Cell-Tak-coated microplate according to the manufacturer’s protocol for non-adherent cells. Platelets were preincubated for 20 min in XF assay medium supplemented with sodium pyruvate (1 mM), L-glutamine (2 mM), and glucose (25 mM) at 37 °C in a CO_2_-free incubator. Oxygen consumption rate (OCR) was subsequently measured every 7 min on an XF96 extracellular flux analyzer (Seahorse Bioscience, North Billerica, MA, USA). A standard protocol was used with a 15 min basal measurement followed by the addition of 2 µM oligomycin (ATP synthase inhibition), 0.2 µM carbonyl cyanide p-trifluoromethoxy-phenylhydrazone (FCCP; proton gradient disruption), and 2.5 µM antimycin A (complex III inhibition). OCR was normalized to protein concentration (pmoles O_2_/(min × µg protein)) using the Pierce^TM^ BCA protein assay kit according to manufacturer’s instructions. Samples were measured as sextuplicate in 5 independent experiments.

### 2.13. In Vitro Thrombus Formation

Vena8Fluoro+ biochips (Cellix, Dublin, Ireland) were coated with collagen (200 µg/mL) overnight at 4 °C and then blocked with BSA (10 µg/mL) for 30 min at RT, followed by two washing steps with dH_2_O. Whole blood collected in 3.8% sodium citrate/ACD was incubated with 3,3-dihexyloxacarbocyanine iodide (1 µM) for 10 min in the dark. As a positive control that inhibits thrombus formation, whole blood was treated with iloprost (10 µM) for 5 min before perfusion. CaCl_2_ (1 mM final concentration) was added to the blood 2 min before perfusion over the collagen-coated chip. Perfusion was performed at a shear rate of 30 dynes/cm^2^. Thrombus formation was recorded every 30 sec for 3 min with a Zeiss Axiovert 40 CFL microscope using a Hamamatsu ORCA-03G digital camera (Hamamatsu, Bridgewater, NJ, USA) and Cellix VenaFlux software. Computerized image analysis was performed using DucoCell analysis software (Cellix), and the area covered by the thrombus was calculated. 

### 2.14. In Vivo Thrombus Formation

Mice were anesthetized with ketamine and xylazine. Before surgery, 60 μL rhodamine 6G (1 mg/mL) (Thermo Fisher Scientific, Waltham, MA, USA) was injected retro-orbitally. After opening the peritoneum, the mesenteric vessels were exposed and thrombus formation was induced by applying a drop of 10% FeCl_3_ [31]. The occlusion time was recorded by intravital microscopy (Olympus Inverted Microscope IX71, equipped with an X-Cite 120PC Q fluorescence lamp, an Olympus TH4-200 halogen lamp, and an Olympus XC50 camera; Tokyo, Japan). 

### 2.15. Statistical Analysis

Statistical analyses were performed using GraphPad Prism 5.0 software. Significance was determined by unpaired Student’s *t* test or ANOVA followed by Bonferroni correction. Data are presented as mean values ± SEM. The following levels of statistical significance were used: * *p* < 0.05, ** *p* ≤ 0.01, *** *p* ≤ 0.001.

## 3. Results

### 3.1. ATGL Is Expressed in Mouse Megakaryocytes and Platelets

To analyze the expression of lipases in mouse megakaryocytes and purified platelets, we performed RT-PCR and Western blotting experiments. We detected mRNA of *Atgl*, *Mgl*, and *Hsl* in megakaryocytes. cDNAs from white adipose tissue (WAT) and macrophages were used as controls (Figure 1A). Additionally, the presence of the ATGL co-activator (*Cgi-58*) and inhibitors (*G0S2* and *Hilpda*) (Figure 1B) showed that the entire lipolytic machinery is expressed in megakaryocytes. In contrast, platelets express *Atgl* and *Mgl* mRNA but not *Hsl* (Figure 1C, lane 1). RNA isolated from whole blood of wt (lane 2) and Atgl−/− mice (lane 3) as well as H_2_O (nCtrl, lane 4) served as controls. Western blot analysis confirmed the presence of ATGL and MGL and the absence of HSL expression in purified mouse platelets (pooled from 6 wt mice) (Figure 1D, lane 1). WAT lysate from a wt mouse (Figure 1C, lane 3) was used as a control. To investigate the purity of our platelet preparations, we analyzed platelet RNA by RT-PCR using primers specific for leukocytes (*Cd45*), erythrocytes (*Cd235a*), and platelets (*Cd41*) (Figure 1E). We used RNA from blood as a positive control. According to mRNA queries in a plateletomics database in 2022 (http://www.plateletomics.com/plateletomics/, accessed on 10 January 2022. Copyright © 2022–2013. Bray Laboratory, Thomas Jefferson University. Shaw Laboratory, Baylor College of Medicine) [32], *ATGL* and *MGL* are among the top 47% and 10% most expressed genes in human platelets, respectively, whereas HSL is much lower expressed (bottom 11%) (Figure 1F). Atgl mRNA is higher expressed in mouse than in human platelets (Appendix A) [32]. 

### 3.2. Little LD Formation in Atgl−/− Platelets

Transmission electron microscopy revealed comparable cellular morphology of platelets from wt and Atgl−/− mice with preserved major organelles such as α-granules (black arrows), dense bodies (white arrows), mitochondria, and the open canalicular system (Figure 2A). Quantification of α-granules and mitochondria on an area of 349 µm^2^ gave a comparable number of these organelles, with 233 versus 222 α-granules and 90 versus 88 mitochondria in wt and Atgl−/− platelets, respectively. The only structural difference we observed was a few LD in Atgl−/− platelets (Figure 2A, lower panel, indicated by asterisk). Quantification revealed that 3% of Atgl−/− platelets had LD. Moreover, scanning electron microscopy demonstrated unchanged morphology in Atgl−/− platelets (Figure 2B). The abundance of a few LD was not associated with significant differences in TG and CE concentrations as determined by UPLC-MS measurements (Figure 2C,D). ATGL was reported to be a cytosolic phospholipase A2 (PLA2) family member, mediating its function via direct release of FA predominantly from phosphatidylcholine (PC), phosphatidylethanolamine (PE), and phosphatidylserine (PS) [33]. The total amounts of PC, PE, and PS were unchanged in Atgl−/− platelets (Appendix A–C), suggesting that ATGL is not an active phospholipase in platelets under unstimulated conditions. However, PL species analysis revealed an increased abundance of PC 32:0, 16:0/16:0, and decreased concentrations of PC 36:4, 16:0/20:4 (Figure 2E). In addition, analysis of PE species revealed a reduced level of PE 38:4, 18:0/20:4 (Appendix A).

### 3.3. Platelet Mitochondrial Respiration Depends on Global Atgl Expression

Since mitochondrial damage or dysfunction markedly impairs platelet function and survival [34] and Atgl−/− macrophages show mitochondrial dysfunction [21,22], we measured oxygen consumption rate (OCR) in platelets from whole-body and platelet-specific (plat) Atgl−/− mice. We detected significantly decreased basal and maximal OCR of Atgl−/− platelets (Figure 3A) but not of platelets from platAtgl−/− mice (Figure 3B). Since mitochondrial respiration is of particular importance for platelet activation [34], we incubated blood from wt, Atgl−/−, Atgl^flox/flox^, and platAtgl−/− mice with various compounds that induce platelet activation, which we measured by analyzing P-selectin expression and integrin αIIbβ3 activation. All agonists tested (ADP, PAR4-AP, CRP, and CVX) showed comparable results between the genotypes (Figure 3C–F), suggesting that platelet activation can be maintained in the absence of ATGL.

### 3.4. Unchanged Hemostatic Function in Atgl−/− Mice

The ability of platelets to form a platelet plug for hemostasis is dependent on platelet aggregation. To assess platelet function, we performed Multiplate^®^ COLtests, which revealed that collagen-induced platelet aggregation is unaffected in Atgl−/− and platAtgl−/− mice (Figure 4A,B). Accordingly, in vivo bleeding assays showed normal bleeding time and blood volume loss in Atgl−/− mice (Figure 4C,D), which was also confirmed by comparable hemoglobin concentrations in isolated blood samples from Atgl−/− and wt mice (Figure 4E).

### 3.5. Loss of ATGL Affects Thrombus Formation In Vitro but Not In Vivo

We next examined platelet reactivity by perfusion of whole blood through collagen-coated channels (Figure 5A) and recorded thrombus formation by fluorescence microscopy (Figure 5B). Iloprost, a structural analogue of prostacyclin, was used as a control to inhibit thrombus formation. The thrombus-covered area was calculated by computerized image analysis and showed markedly reduced in vitro thrombus formation after 2, 2.5, and 3 min of perfusion in blood from Atgl−/− mice (Figure 5C) despite unaltered number (data not shown) and volume of platelets between wt and Atgl−/− mice irrespective of sex (Figure 5D).

Collagen is able to activate platelets upon surface contact and requires the presence of the collagen receptor glycoprotein VI (GPVI) on the platelet membrane [35]. Based on the reduced thrombus formation observed in collagen-coated channels, we determined the expression level of GPVI in resting and CRP-stimulated cells. The surface expression and shedding of GPVI [36] in response to CRP was comparable between the genotypes (Figure 5E). Thromboxane B2 (TXB2), the stable metabolite of thromboxane A2, is produced by activated platelets during platelet plug formation and has pro-thrombotic properties [37]. Thus, we investigated whether the attenuated thrombus formation was the result of decreased TXB2 levels, since ATGL deficiency may cause reduced availability of FA for lipid mediator production [24]. However, the plasma TXB2 concentrations were unaltered in Atgl−/− mice (Figure 5F). Finally, we assessed the occlusion time of mesenteric vessels in vivo by intravital microscopy using the FeCl_3_-induced thrombus formation model [31]. Occlusion time was comparable between Atgl−/− and control mice (Figure 5G).

## 4. Discussion

Platelets are capable of synthesizing FA and PL de novo [38,39]; however, it remained elusive as to whether the lipolytic machinery in these cells is present and contributes to FA homeostasis. In most cell types, neutral lipolysis of cytosolic LD represents an important biochemical mechanism for the release of FA, which are further used as energy substrates, precursors for lipid and membrane synthesis, and ligands for various signaling processes.

Based on our observations, mouse megakaryocytes express *Atgl*, *Hsl*, and *Mgl* mRNA. Mouse platelets, however, express only ATGL and MGL but not HSL protein. Consistent with these data, genome-wide RNA sequencing showed that *Mgl* and *Atgl* are highly expressed in human platelets, whereas *Hsl* is expressed only at low levels [26]. In contrast to typical LD-storing cells (e.g., adipocytes or macrophages), lipids in platelets are primarily components of membranes. PL are the major structural lipids (~75%) of the platelet membrane bilayer and provide the substrates for the formation of bioactive lipid mediators [40,41]. Neutral lipids and cholesterol account for the remaining 25%, with cholesterol being the predominant species [41]. Activated platelets also contain a considerable amount of FA, of which arachidonic acid is the most important as a precursor for oxidative transformation to several eicosanoids by lipoxygenase and cyclooxygenase [4]. Due to the absence of HSL, platelets express only a partially preserved classical lipolytic machinery, and the additional loss of ATGL resulted in the accumulation of only few LD in this cell type, arguing against an important physiological relevance. ATGL has also been attributed a functional role as a phospholipase or transacylase [33]. The canonical pathway of eicosanoid production in platelets is initiated by phospholipase A_2_-mediated release of arachidonic acid [42]. The rearrangement of FA in PL is an important process during platelet activation, and several acyltransferases for the acylation of lysophospholipids have been detected in platelets [43,44]. Unchanged total PE, PC, and PS levels between wt and Atgl−/− platelets suggest that ATGL is of minor importance as a phospholipase in resting platelets. However, PC 32:0, 16:0/16:0 was selectively increased in Atgl−/− platelets. Incubation of platelets with PC 32:0, 16:0/16:0 was shown to increase platelet PC levels, suppress platelet adhesion, and reduce the number of δ-granules released per secretion event [45].

In contrast to Atgl−/− macrophages, which exhibit mitochondrial dysfunction resulting in smaller and fragmented mitochondria [22], loss of ATGL has no effect on the morphology of platelet mitochondria. Of note, Atgl−/− macrophages accumulate drastically more TG, which we did not observe in Atgl−/− platelets and may be due to a lower number of LD. However, basal and maximal mitochondrial respiration of platelets from whole-body Atgl−/− mice were significantly decreased, whereas OCR of platelets from platAtgl−/− mice was unchanged, indicating that platelet-specific ATGL deficiency is not responsible for the observed changes in mitochondrial function. One explanation for the alterations in mitochondrial respiration might be that energy supply is insufficient due to decreased FA availability in the plasma of global Atgl−/− mice [17]. Moreover, it was shown that FA can be transported in neutrophil-derived extracellular vesicles, which are then internalized into platelets [46]. Since ATGL regulates FA availability in myeloid cells [24], FA transport in neutrophil-derived extracellular vesicles might be impaired. In platAtgl−/− mice, however, FA can still be provided by neutrophils, monocytes and macrophages. Despite reduced OCR in Atgl−/− platelets, platelet activation, aggregation, and hemostatic function were unaffected. Oxidative phosphorylation provides 30–40% of cellular ATP, but most of the energy is derived from glycolysis [47]. Several key processes that occur in platelets require a constant energy supply. Platelets show metabolic flexibility that helps them meet these energy needs by utilizing either glycolysis or mitochondrial ATP production to adapt to different situations [48].

Changes in plasma lipoprotein composition have an important effect on platelet function. Whole-body ATGL deficiency is associated with systemic changes in lipid metabolism, which are reflected in altered lipid concentrations and lipoprotein profiles. Plasma concentrations of cholesterol, TG, and FA are markedly reduced, and lipoprotein profiles show decreased VLDL, LDL, and HDL fractions [18]. The lipid composition of membranes determines membrane-mediated platelet activities such as membrane fluidity, eicosanoid generation, and signaling pathways and is influenced by various factors. The beneficial anti-aggregative and anti-thrombotic effects of statins [12,49,50] may be due to their influence on platelet activity by lowering the cholesterol content in the platelet plasma membrane [51]. In vitro thrombus formation assays were significantly reduced in Atgl−/− blood samples. Comparable platelet counts and surface expression levels of GPVI, which activates platelets by collagen [52], exclude the possibility that alterations in platelet numbers or collagen receptor expression are responsible for the reduced thrombotic phenotype of Atgl−/− blood. One possible explanation for reduced in vitro thrombus formation might be hypolipidemia caused by a systemic deficiency of ATGL. Of note, Atgl−/− neutrophils show reduced release of arachidonic acid and consequently TXB2, as well as many other lipid mediators [24], which are important for platelet activation. Similar to the anti-inflammatory effect of lipid-lowering drugs such as statins in the vasculature that prevents thrombotic events [50], loss of ATGL polarizes macrophages toward an anti-inflammatory M2-like phenotype [20]. Since platelets are able to sequester cytokines released by monocytes [53], the anti-inflammatory state of Atgl−/− monocytes could protect from collagen-induced thrombus formation. Despite the decreased in vitro thrombus formation in Atgl−/− blood, vessel occlusion in the FeCl_3_ injury model was comparable between Atgl−/− and control animals. The in vitro thrombosis assay detects thrombosis through the collagen-related pathway, whereas vessel occlusion in vivo is based on the FeCl_3_-induced oxidative damage of vascular cells with a highly complex sequence of various events (such as platelet adhesion, activation, aggregation, granule release, and coagulation activity), leading to a fibrin-rich thrombus formation, and may fail to detect subtle but significant effects on mechanistic aspects of thrombosis [54]. In addition, a mild defect might not be evident in this model because, in contrast to the in vitro thrombus formation assay, plasmatic hemostasis (tissue factor-induced thrombin generation) and endothelial cells might also play a compensatory role [31].

In summary, the present study demonstrates that decreased mitochondrial respiration of platelets is associated with reduced collagen-induced thrombus formation in vitro without pathological bleeding and platelet activation in the absence of ATGL in vivo.

## Figures and Tables

**Figure 1 cells-11-00850-f001:**
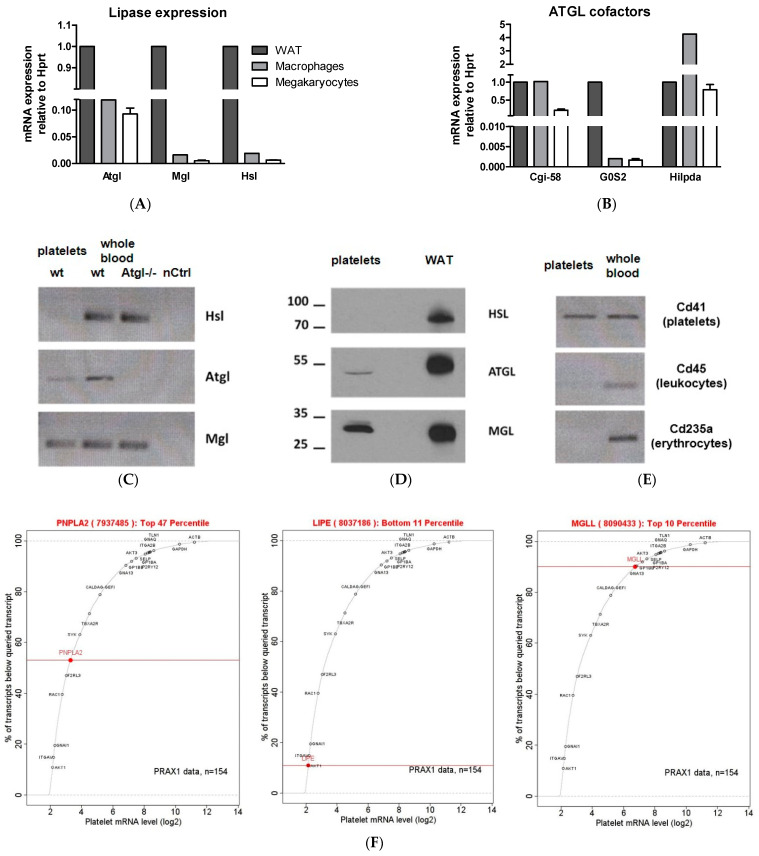
Neutral lipases are expressed in mouse and human platelets. mRNA expression of (**A**) adipose triglyceride lipase (*Atgl*), monoglyceride lipase (*Mgl*), hormone-sensitive lipase (*Hsl*), and (**B**) ATGL cofactors in megakaryocytes (*n* = 6) relative to the expression of hypoxanthine phosphoribosyltransferase 1 (*Hprt*) as a housekeeping gene. (**C**) RT-PCR of neutral lipases in purified platelets (lane 1). Whole blood RNA from wt (lane 2) and Atgl−/− mice (lane 3) as well as ddH_2_O (lane 4) were used as controls. (**D**) Lysates of platelets (pooled from 6 wt mice) were separated by SDS-Page and analyzed for HSL, ATGL, and MGL protein expression (lane 1) by Western blotting. Protein lysate of white adipose tissue (WAT) from a wt mouse was used as a positive control. (**E**) To confirm platelet purity, platelet RNA was analyzed with primers specific for platelets (*Cd41*), leukocytes (*Cd45*), and erythrocytes (*Cd235a*). (**F**) Human platelet mRNA expression data from 154 healthy subjects determined by microarray (http://www.plateletomics.org/plateletomics/, accessed on 10 January 2022, Copyright © 2022–2013. Bray Laboratory, Thomas Jefferson University. Shaw Laboratory, Baylor College of Medicine) [32] revealed mRNA expression of human ATGL (*PNPLA2*), HSL (*LIPE*), and MGL (*MGLL*) relative to the expression of reference genes.

**Figure 2 cells-11-00850-f002:**
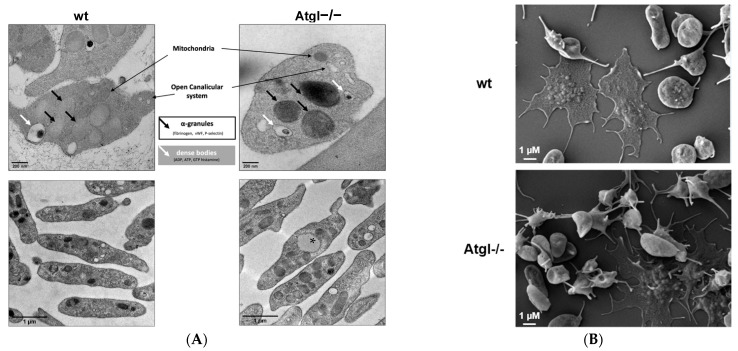
Sporadic lipid droplet formation but unchanged overall morphology in Atgl−/− platelets. (**A**) Representative electron micrographs of platelets from wt and Atgl−/− mice. The asterisk indicates a cytosolic LD. Scale bar, 200 nm (upper panel) and 1 µm (lower panel). (**B**) Platelet spreading was visualized by scanning electron microscopy. Scale bar, 1 µM. Lipids were extracted from platelets (pooled from 6 mice) and analyzed by UPLC-MS to quantify (**C**) triglyceride (TG) and (**D**) cholesteryl ester (CE) concentrations and determine (**E**) phosphatidylcholine (PC) species. Data are shown as mean + SEM (*n* = 5). ***, *p* < 0.001.

**Figure 3 cells-11-00850-f003:**
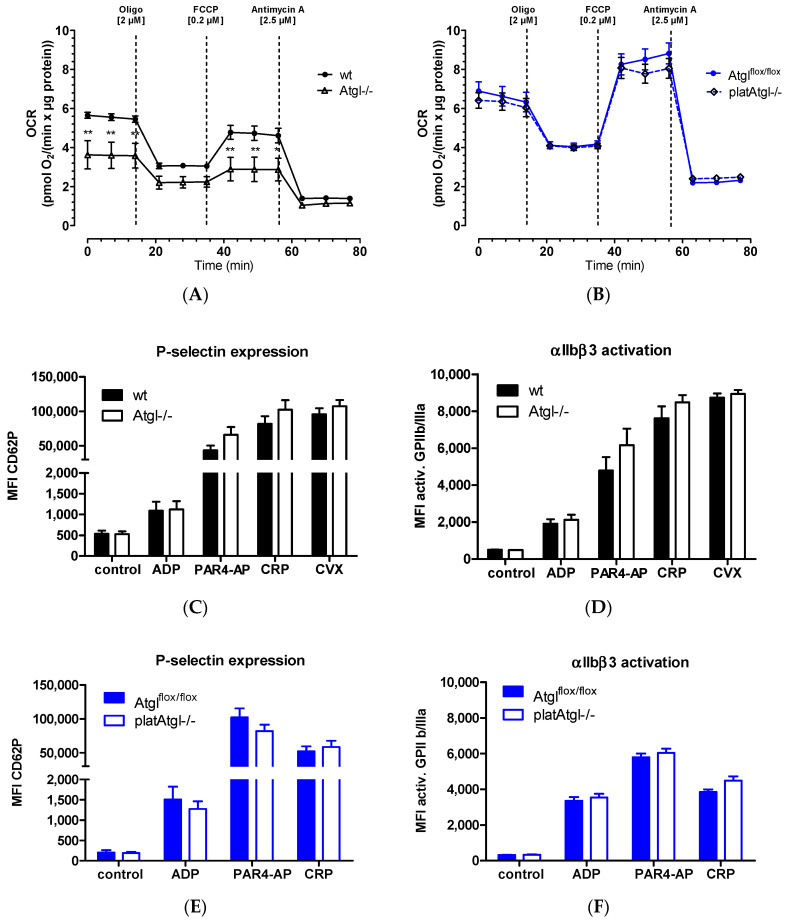
Reduced mitochondrial respiration in whole-body Atgl−/− mice. Oxygen consumption rate (OCR) in isolated platelets from (**A**) Atgl−/− and (**B**) platAtgl−/− mice and their respective controls determined on a Seahorse XF Analyzer. Ten million cells were seeded in XF assay medium supplemented with sodium pyruvate (1 mM), L-glutamine (2 mM), and glucose (25 mM) per 96-well. Cells were treated with 2 μM oligomycin (Oligo), 0.2 μM carbonyl cyanide-4-(trifluoromethoxy)phenylhydrazone (FCCP), and 2.5 μM antimycin A. Values were normalized to protein content using the PierceTM BCA protein assay kit according to manufacturer’s instructions. Data are presented as mean values ± SEM of sextuplicate from 5 independent experiments. *, *p* < 0.05; **, *p* < 0.01. Significance was calculated by ANOVA followed by Bonferroni post hoc test. Blood from (**C**,**D**) Atgl−/− and (**E**,**F**) platAtgl−/− mice was activated with ADP (50 µM), protease-activated receptor 4 agonist peptide (PAR4-AP, 75 µM), collagen-related peptide (CRP, 10 µg/mL), and convulxin (CVX, 125 ng/mL) in the presence of PE-Cy7-conjugated anti-mouse P-selectin antibody and JON/A-PE antibody directed against the activated form of integrin αIIbβ3. Data are shown as geometric mean of fluorescence intensity (MFI) + SEM (*n* = 7–8).

**Figure 4 cells-11-00850-f004:**
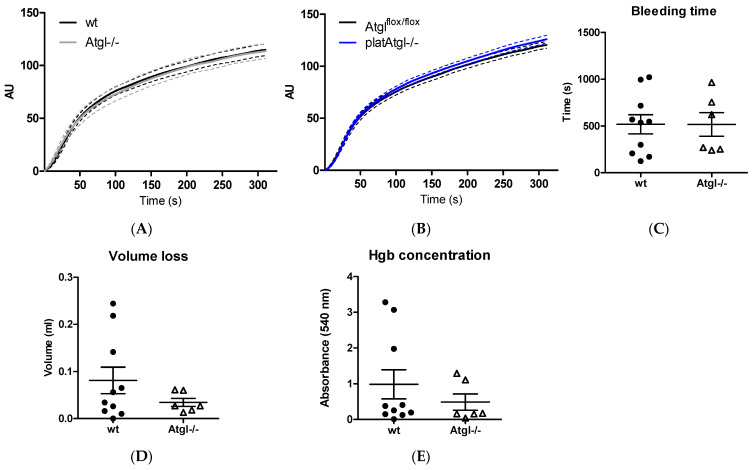
Unchanged hemostatic function in Atgl−/− mice. Platelet aggregation in (**A**) Atgl−/− and (**B**) platAtgl−/− blood was measured using a Multiplate^®^ analyzer. Data are expressed as mean arbitrary units (AU) (*n* = 10). (**C**) Bleeding time, (**D**) blood volume loss, and (**E**) hemoglobin (Hgb) concentration as absorbance at 540 nm were determined in isolated blood of Atgl−/− mice. Data are shown as mean ± SEM (*n* = 6–10).

**Figure 5 cells-11-00850-f005:**
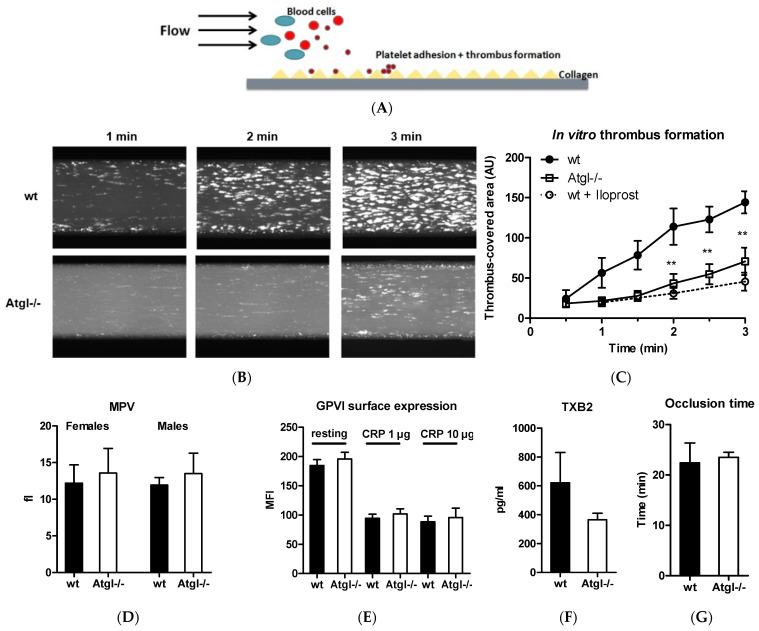
Reduced *in vitro* thrombus formation in Atgl−/− blood. (**A**) Platelet reactivity was determined by in vitro thrombus formation, in which platelets were stained in whole blood and perfused over collagen-coated channels. (**B**) In vitro thrombus formation of wt and Atgl−/− blood every 30 sec up to 3 min of perfusion was recorded by fluorescence microscopy. (**C**) The thrombus-covered area was calculated by computerized image analysis. Iloprost, which inhibits thrombus formation in wt blood, was used as a positive control. Data are expressed in arbitrary units (AU) (mean ± SEM) (*n* = 10). ** *p* ≤ 0.01. (**D**) Mean platelet volume (MPV) from whole blood was determined by automated cell counting. Data represent mean values + SEM (*n* = 3–9). (**E**) Whole blood (resting and CRP-stimulated) was stained with an FITC-conjugated anti-GPVI antibody, and platelets were analyzed by flow cytometry. Data are shown as geometric mean of fluorescence intensity (MFI) + SEM (*n* = 5). (**F**) Plasma TXB2 concentrations were measured by ELISA (mean + SEM) (*n* = 6). (**G**) Thrombus formation was induced by applying a drop of 10% FeCl_3,_ and occlusion time was recorded by intravital microscopy. Data are presented as mean + SEM (*n* = 5–6).

## Data Availability

The data presented in this study are available on reasonable request from the corresponding author. Reagents and detailed methods of all procedures are provided in Section 2 of this manuscript or cited accordingly.

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
