# Peer review of "Adipose Triglyceride Lipase Deficiency Attenuates In Vitro Thrombus Formation without Affecting Platelet Activation and Bleeding In Vivo"

_cells, 2022, doi:10.3390/cells11050850_

Round 1

Reviewer 1 Report

  The paper by Goeritzer et al. aimed to investigate how the loss of ATGL on platelet function. This is a well-designed and prepared study and the reviewer does not have concerns regarding the manuscript and it is ready for publication. 

Author Response

We would like to thank the Reviewer for the very positive comments on our manuscript.

Reviewer 2 Report

In this manuscript Goeritzer et al investigated the adipose triglyceride lipase deficiency in platelet function and found a decreased in vitro thrombus formation but no other major platelet dysfunction. These are my comments:

Major:

The method description does not always contain relevant information regarding cell count. It would be important to indicate the platelet count for the spreading, flow cytometry, aggregation.

In the method description the authors state that “to visualize platelet spreading washed platelet suspensions were placed on collagen- and alcian blue coated cover slips and immediately fixed…”. If there was no incubation of the platelets on the collagen surface to allow them to spread, it cannot be called spreading. The representative images shown in Figure 2B are not spread platelets, but rather platelet clumps fixed to the surface. If spreading would occur, various state of spreading should be seen such as filopodia and lamellopodia formation. What is the definition of spreading according to the authors? Furthermore, based on these images, the authors concluded that Atgl-/- platelets were smaller, and it may seem indeed that the wild type platelets are larger, but this should be concluded based on MPV values and not on these images. In the current setup the size may vary due to the different degree of adhesion to the surface. Furthermore, the more filopodia like structure is seen with the Atgl-/- platelets which may be suggestive for a higher (pre)activation state of the platelets. Did the authors encounter any problems regarding pre-activation during this experiment?

The platelet morphology did not differ except for the occasional lipid droplets in the Atgl-/- platelets based on EM. It is likely that the representative images are not the best in Figure 2A, but it seems that the number or organelles (e.g. alpha granules) are different. Have the authors quantified those? It would be helpful to report. Regarding the occasional LD in some platelets, what was the fraction of these platelets?

Platelet reactivity was measured by P-selectin and integrin expression upon various stimuli and no difference was found between both types of knock out mice and the corresponding controls. There seem to be an opposite trend in P-selectin expression upon PAR4 stimulation. Only a high concentration of agonist was used, but it is possible these concentrations are overshooting the limit. Have the authors tried various concentration of the agonists to see whether suboptimal activation would give a different outcome?

When the platelet function was assessed by Mulitplate assay, no difference occurred between the KO and WT mice. In the figure legend (figure 4), n=10 is stated for this experiment, but only one representative curve is shown. Please add quantification of these data to show the variability between experiments which could be suggestive for further interpretation.

Lastly, platelet reactivity was investigated in a microfluidic system. It is described that “CaCl2 was spread over the collagen coated chip 2 min before perfusion…” What was the rational behind it? Was it washed away before the blood entered the channel? It is not enough to fully re-calcify the whole blood and it will be washed away during blood perfusion.  

The authors suggest that ATGL inhibition may be a good alternative for preventing collagen-induced thrombotic events without further increase in bleeding risk. However, the in vivo result does not support such a conclusion. One of the arguments that the authors provide for the discrepancy between the in vivo and in vitro data, is that the in vitro assay depends solely on collagen, whilst the FeCl3 model not. But it has been shown by a detailed meta-analysis including appr. 1500 published studies that collagen dependent microfluidic assays have a good correlation with the FeCl3 model. What else could be the explanation behind the discrepancy between the two assays? Also, in which pathological condition could a ATGL inhibitor be used? Do the authors have any prove that ATGL inhibition results in the same outcome in humans as well?

Minor:

  1. I would suggest indicating the full name of ATGL in the title instead of the abbreviation.
  2. The manuscript contains several small mistakes such as not consistent axis labels (e.g. Par4 vs PAR4-AP). These should be corrected.

Round 2

Reviewer 2 Report

In the revised version, Goeritzer et al adequately answered most of my concerns; however, I have one question which remained unanswered.

It is stated that some Atgl-/- platelets accumulated lipid droplets. What was the fraction of platelets containing LD compared to normal platelets? It is quite vague in the current from without any quantification and it makes it harder to see whether or not it has a real physiological relevance
